# Female sexual function index for screening of female sexual dysfunction using DSM-5-TR criteria in Thai women: A prospective cross-sectional diagnostic study

**Patthamaphorn Chongcharoen**®, **Thanapan Choobun, Siwatchaya Khanuengkitkong**® *

Department of Obstetrics & Gynecology, Faculty of Medicine, Prince of Songkla University, Hat Yai, Songkhla, Thailand

* meawpetch@yahoo.com

**Data Availability Statement:** All relevant data are within the manuscript and its Supporting Information files.

## Abstract

Thai Female Sexual Function Index discrimination using the new Diagnostic and Statistical Manual of Mental Disorders, Fifth Edition, Text Revision criteria has not been investigated. This study aimed to evaluate the Female Sexual Function Index as a tool for assessing sexual symptoms and to determine the prevalence of female sexual dysfunction in Thai women using the new Diagnostic and Statistical Manual of Mental Disorders, Fifth Edition, Text Revision criteria. This prospective cross-sectional diagnostic study included sexually active women aged ≥18 years, interviewed from January to June 2023. The participants completed the Thai version of a comprehensive of the Female Sexual Function Index questionnaire encompassing general information and self-reported assessments of female sexual function, followed by a semi-structured interview of distress symptom severity. Female sexual function was determined by screening of the total Female Sexual Function Index score, whereas female sexual dysfunction was evaluated using the Diagnostic and Statistical Manual of Mental Disorders, Fifth Edition, Text Revision criteria. Using receiver operating characteristic curves, a clinical cutoff for the Female Sexual Function Index score of 23.1 was determined to identify female sexual dysfunction (area under the curve, 0.76; 95% confidence interval, 0.71–0.80; sensitivity, 75.6%; specificity, 67.7%; positive predictive value, 77.7%; negative predictive value, 65%). A prevalence of 40.2% for female sexual dysfunction was observed in the study population. The results of this study could be used as practical guidance for the screening of women affected by female sexual dysfunction in Thailand in the future.

## Introduction

Female sexuality is a fundamental human right and an important aspect of women's health. The World Health Organization defines sexual health as physical, emotional, mental, and social well-being related to sexuality rather than the absence of dysfunction or disease [1].

**Funding:** This research received financial support from the Faculty of Medicine, Prince of Songkla University, through Grant 65-086-1, with the Primary Investigator being Patthamaphorn Chongcharoen. It is important to note that the funders had no role in the study design, data collection and analysis, decision to publish, or manuscript preparation.

**Competing interests:** The authors report there are no conflicts of interest to declare.

Female sexual dysfunction (FSD) and other sexual problems have been associated with a diminished quality of life, low physical and emotional satisfaction, and reduced general happiness [2].

Sexual dysfunction refers to pain during sexual intercourse or disturbances in one or more phases of the sexual response cycle, and is a concern in family life. It can have disastrous effects on mental well-being, sense of worth, social acceptance, and interpersonal relationships, as well as cause marital problems [3]. The worldwide prevalence of FSD has been reported to be between 30% and 70% [4, 5], whereas in Asian countries such as China, Korea, and Japan, it has been reported to be in the 30%–40% range [6, 7].

The Diagnostic and Statistical Manual of Mental Disorders, (Fifth Edition) (DSM-5) [3], the gold standard for diagnosis of FSD, identifies four specific types of sexual dysfunction in women, including a spectrum of disorders that are typically multifactorial in etiology, such as sexual interest/arousal disorder (SIAD), female orgasmic disorder (FOD), and genito-pelvic penetration pain disorder (GPPPD). Additionally, it classifies disorders as substance/medication-induced, other specified, and unspecified sexual dysfunctions.

Sexual responses do not always follow a steady or uniformly linear pattern; therefore, labeling distinct phases such as desire and arousal as separate entities might not truly reflect their nature [8]. The DSM-5-TR combines sexual desire and arousal disorders in women into a female sexual interest/arousal disorder (FSAID) [3, 4]. Furthermore, to refine the precision related to duration and severity criteria and reduce the risk of excessive diagnosis, the DSM-5-TR requires approximately 6 months as the minimum duration, along with more meticulous guidelines, to assess the severity for sexual dysfunction.

To date, the Female Sexual Function Index (FSFI) questionnaire [9], a self-reported patient screening tool, scores symptoms against normative values for women with and without sexual dysfunction. A Thai version of the questionnaire [10] has been validated as a practical method for obtaining professional advice and knowledge on sexuality, leading to timely and effective treatment. However, the original FSFI [9, 11] differentiated FSD based on the Diagnostic and Statistical Manual of Mental Disorders, Fourth Edition, Text Revision (DSM-IV-TR) criteria. Therefore, the discriminative capacity of the Thai FSFI with the new DSM-5-TR criteria has not been investigated. The primary objective of the present study was to define the optimal Thai FSFI cutoff point for FSD screening using DSM-5-TR criteria in Thai women. The secondary objective was to determine the prevalence of FSD in Thai women using DSM-5-TR criteria.

## Materials and methods

This prospective cross-sectional diagnostic study included sexually active women and was conducted at [blinded for peer-review] Hospital following approval from the research ethics committee (REC no. 65-394-12-1). The procedures adhered to the ethical standards of the Helsinki Declaration (1964). The participants were recruited between 1 January and 30 June 2023, through posters in the outpatient clinic ([blinded for peer review]). We also recruited participants in [blinded for peer review] province after explaining that the study protocol involved women's sexual health. The inclusion criteria comprised heterosexual women aged $\geq$ 18 years engaged in sexually active relationships. Exclusion criteria included pregnancy, breastfeeding, being sexually inactive, having a partner with sexual dysfunction, refusing to participate, or being unable to read. Written informed consent was obtained from all participants after providing explanations about of the methods involved in assessing women's sexual health.

The study tools and outcome measurements consisted of two parts. Part 1 involved participants completing a questionnaire in Thai language comprising 30 questions. Questions 1–9

assessed demographic data and general health. Question 10 assessed the cohabitation status of the participant's sexual partner, and the frequency of sexual activity. Question 11 focused on self-reported FSD, and questions 12–30 comprised the Thai version of the FSFI, (self-reported measures of female sexual function) [10]. The FSFI provides scores for overall levels of sexual function [9, 12]. Each question in the FSFI was assigned a score range of 0–5. The sum of each domain score was first multiplied by a domain factor ratio and added to derive the total FSFI score. Lower scores indicate a higher degree of sexual dysfunction. The participants were allowed to complete the questionnaire independently after receiving written informed consent.

In part 2 of the study, the participants underwent a semi-structured interview about their overall distress. Each difficulty was defined by the DSM-5-TR criteria as having symptoms that last > 6 months, experiencing symptom severity levels > 75%–100%, reporting moderate to severe distress that affects the person and their partner, and possibly needing to avoid sexual activity. The one to one interview was conducted by a standardized physician over the age of 30 (urogynecologists and fellows trained in female pelvic medicine and reconstructive surgery) to determine whether or not they met the DSM-5-TR criteria for any of the following disorders: FSIAD, FOD, or GPPPD. FSD can be a single, two-, or three-dimensional disorder. The interview was conducted in a quiet and private setting. This ensured physical and mental comfort, safety, anonymity, solitude, freedom from judgment, and the ability for the participants to express themselves.

The sample size was determined based on estimating a single proportion test [13, 14] using the 90% sensitivity of the FSFI score for sexual dysfunction. A study with a power of 80%, a type I error of 5%, and an increased data loss of 10% required an estimated sample size of 155 participants. Assuming an estimated FSD prevalence of 0.31 [15], 500 participants were required. Statistical analyses were conducted using R-Software version 4.2.3 [16]. Continuous variables are presented as mean ± SD or median (IQR), and discrete variables are presented as numbers (percentage). A one-sample t-test was used to calculate domain scores and FSD ratios. A P value < 0.05 was considered significant. To establish clinical cutoff scores for the Thai version of the FSFI, a Receiver Operating Characteristic (ROC) curve was constructed. The optimal cutoff score was defined as the one that minimized the distance between the ROC curve and the upper left corner.

## Results

In total, 500 participants (median age, 39 years; IQR, 33–46 years) were enrolled. Among them, 80% reported good physical health, 74% had children, and 61.6% used contraceptives regularly. The majority of participants (81.4%) were premenopausal, and 27% reported engaging in sexual intercourse 2–3 times per month. Furthermore, 71% indicated normal self-sexual function, whereas 29% expressed an attitude toward having self-sexual dysfunction, and 60.7% reported a desire disorder. Problems with lubrication, pain, arousal, orgasm, and satisfaction were reported at rates of 30.3%, 28.9%, 20%, 15.9%, and 11.7%, respectively. Attitudes toward having self-sexual dysfunction were highest in the 40–49-year-old age range (47 individuals, 32.5%). In this age group, FSD was more common in at least one domain.

Participants in the FSD group (n = 201) were older (42 vs. 38 years, P<0.001). The group also had a higher proportion of women aged >50 years (26.9% vs. 10%, P<0.001) and in postmenopausal status (28.9% vs. 11.7%, P<0.001), more women with a frequency of sexual activity of once per month (19.4% vs. 13%, P<0.001), and a more negative attitude toward self-sexual dysfunction (42% vs. 20.1%, P<0.001) (Table 1).

**Table 1. Demographic characteristics of the study population (N = 500).**

| Characteristic/Measurement | Women without FSD (n = 299) n (%)[a] | Women with FSD (n = 201) n (%)[b] | Total n (%) | P value |
|---|---|---|---|---|
| Age, years: median (IQR) | 38 (32.5,44) | 42 (35,50) | 39 (33,46) | **<0.001** |
| Age group | | | | **<0.001** |
| <29 years | 48 (16.1) | 28 (13.9) | 76 (15.2) | |
| 30–39 years | 124 (41.5) | 56 (27.9) | 180 (36) | |
| 40–49 years | 95 (31.8) | 60 (29.9) | 155 (31) | |
| 50–59 years | 30 (10) | 54 (26.9) | 84 (16.8) | |
| ≥60 years | 2 (0.7) | 3 (1.5) | 5 (1) | |
| Menopausal status | | | | **<0.001** |
| Premenopause | 264 (88.3) | 143 (71.1) | 407 (81.4) | |
| Postmenopause | 35 (11.7) | 58 (28.9) | 93 (18.6) | |
| BMI, kg/m$^2$ | | | | |
| Underweight (<18.5) | 15 (5) | 8 (4) | 23 (4.6) | 0.296 |
| Normal (18.5–22.9) | 125 (41.8) | 72 (35.8) | 197 (39.4) | |
| Overweight and obese (>23) | 159 (53.2) | 121 (60.2) | 280 (56) | |
| Educational level | | | | 0.06 |
| None | 1 (0.3) | 1 (0.5) | 2 (0.4) | |
| Primary school | 16 (5.4) | 9 (4.5) | 25 (5) | |
| Secondary school | 116 (38.8) | 101 (50.2) | 217 (43.4) | |
| College and university | 166 (55.5) | 90 (44.8) | 256 (51.2) | |
| Occupation | | | | 0.974 |
| Housewife | 5 (2.6) | 3 (1.5) | 8 (1.6) | |
| Government employer | 79 (41.6) | 49 (42.6) | 128 (42) | |
| Private employee & others | 106 (55.8) | 63 (54.8) | 169 (55.4) | |
| Salary (baht) | | | | 0.256 |
| <10,000 | 45 (15.2) | 20 (10.1) | 65 (13.2) | |
| 10,000–30,000 | 55 (18.6) | 40 (20.2) | 95 (19.2) | |
| >30,000–39,999 | 196 (66.2) | 138 (69.7) | 334 (67.6) | |
| Marital status | | | | 0.322 |
| Never married | 36 (12) | 22 (10.9) | 58 (11.6) | |
| Married | 250 (83.6) | 164 (81.6) | 414 (82.8) | |
| Divorced, separate, widowed | 13 (4.3) | 15 (7.5) | 28 (5.6) | |
| Parity | | | | 0.489 |
| None | 79 (26.4) | 50 (24.9) | 129 (25.8) | |
| 1 | 87 (29.1) | 51 (25.4) | 138 (27.6) | |
| >1 | 133 (44.5) | 100 (49.8) | 233 (46.6) | |
| Health status | | | | 0.599 |
| Healthy/no underlying disease | 238 (83.8) | 152 (79.6) | 390 (82.1) | |
| Hypertension | 10 (3.5) | 5 (2.6) | 15 (3.2) | |
| Diabetes Mellitus | 3 (1.1) | 5 (2.6) | 5 (1.1) | |
| Dyslipidemia | 16 (5.6) | 15 (7.9) | 31 (6.5) | |
| Other | 17 (6) | 17 (8.9) | 34 (7.2) | |
| Non-smoker or alcohol or illicit drugs user | 277 (92.6) | 191 (95) | 468 (93.6) | 0.378 |
| Current smoker or alcohol or illicit drug user | 22 (7.4) | 10 (5) | 32 (6.4) | |
| Current medication | | | | 0.431 |
| Antihypertensive use | 16 (55.2) | 15 (68.2) | 31 (60.8) | |
| Anxiolytic use | 4 (13.7) | 2 (9.1) | 6 (11.7) | |
| Antidepressant use | 4 (13.7) | 3 (13.6) | 7 (13.7) | |

*(Continued)*

**Table 1.** (Continued)

| Characteristic/Measurement | Women without FSD (n = 299) n (%)[a] | Women with FSD (n = 201) n (%)[b] | Total n (%) | P value |
|---|---|---|---|---|
| Hormonal replacement use | 3 (10.4) | 0 (0) | 3 (5.9) | |
| Hypnotic agent use | 2 (7) | 2 (9.1) | 4 (7.9) | |
| Contraception method | | | | 0.119 |
| Nonuser | 106 (35.5) | 86 (42.8) | 192 (38.4) | |
| Used at least 1 method | 193 (64.5) | 115 (57.2) | 308 (61.6) | |
| Feeling depressed/distressed this month | | | | 0.653 |
| Present | 25 (8.4) | 20 (10) | 45 (9) | |
| Absent | 274 (91.6) | 181 (90) | 455 (91) | |
| Partner living together (couple relationship) | 240 (80.3) | 149 (74.1) | 389 (77.8) | 0.131 |
| Frequency of sexual activity | | | | <**0.001** |
| >3 times per week | 24 (8) | 13 (6.5) | 37 (7.4) | |
| 2–3 times per week | 87 (29.1) | 33 (16.4) | 120 (24) | |
| 1 time per week | 50 (16.7) | 28 (13.9) | 78 (15.6) | |
| 2–3 times per month | 87 (29.1) | 48 (23.9) | 135 (27) | |
| 1 time per month | 39 (13) | 39 (19.4) | 78 (15.6) | |
| 1 time per 6 months | 7 (2.3) | 22 (10.9) | 29 (5.8) | |
| 1 time per year | 5 (1.7) | 18 (9) | 23 (4.6) | |
| Sexual orientation | | | | 0.215 |
| Heterosexual | 276 (92.3) | 182 (90.5) | 458 (91.6) | |
| Homosexual | 10 (3.3) | 6 (3) | 16 (3.2) | |
| Bisexual | 0 (0) | 1 (0.5) | 1 (0.2) | |
| Polysexual | 4 (1.3) | 1 (0.5) | 5 (1) | |
| Omnisexual | 2 (0.7) | 0 (0) | 2 (0.4) | |
| Asexual | 7 (2.3) | 11 (5.5) | 18 (3.6) | |
| Attitude toward having self-sexual dysfunction | | | | <**0.001** |
| Present | 60 (20.1) | 85 (42.3) | 145 (29) | |
| Absent | 239 (79.9) | 116 (57.7) | 355 (71) | |

FSD, female sexual dysfunction; IQR, interquartile range; BMI, body mass index.

[a,b] P value for comparison.

The prevalence of FSD, according to the DSM-5-TR criteria, was 40.2% (Table 2). GPPPD, affecting 142 women (70.6%), was the most common subtype of FSD, followed by FOD and FSAID. Furthermore, 67 women (33.3%) had single-dimensional FSD, whereas 72 (35.8%) and 62 women (30.9%) had two- and three-dimensional FSD, respectively.

**Table 2. Female sexual dysfunction classified by age group (n = 201).**

| Age group | FSAID* n (%) | FOD* n (%) | GPPPD* n (%) | One FSD n (%) | Two FSD n (%) | Three FSD n (%) | Any FSD n (%) |
|---|---|---|---|---|---|---|---|
| <29 years | 13 (10.8) | 18 (13.3) | 23 (16.2) | 10 (14.9) | 10 (13.9) | 8 (12.9) | 28 (13.9) |
| 30–39 years | 29 (24.2) | 36 (26.7) | 40 (28.2) | 22 (32.8) | 19 (26.4) | 15 (24.2) | 56 (27.9) |
| 40–49 years | 35 (29.2) | 41 (30.4) | 39 (27.5) | 19 (28.4) | 27 (37.5) | 14 (22.6) | 60 (29.9) |
| 50–59 years | 42 (35.0) | 39 (28.9) | 38 (26.8) | 14 (20.9) | 15 (20.8) | 25 (40.3) | 54 (26.9) |
| ≥60 years | 1 (0.8) | 1 (0.7) | 2 (1.4) | 2 (3.0) | 1 (1.4) | 0 (0) | 3 (1.5) |
| Total (n) | 120 | 135 | 142 | 67 | 72 | 62 | 201 |

* Participants with sexual dysfunction in more than one domain.

FSAID, female sexual interest/arousal disorder; FOD, female orgasmic disorder; GPPPD, genito-pelvic penetration pain disorder; FSD, female sexual dysfunction.

The median overall FSFI score and each of the six domains are shown in Table 3. The median FSFI score was 24.5 (IQR 20.2–28). Both the total score and the scores for each domain showed significant differences between the FSD and non-FSD groups, with lower scores observed in the FSD group (P<0.001).

We also observed that scores below 23.1 indicated FSD based on the DSM-5-TR criteria. This cutoff point exhibited a sensitivity of 75.6%, specificity of 67.7%, positive predictive value (PPV) of 77.7%, and negative predictive value (NPV) of 65%. We conducted subgroup analyses based on cutoff values derived from the total FSFI score and various characteristics. Our findings revealed that among women aged over 40 years, the optimal cutoff value was 24.1, demonstrating a sensitivity of 59.3% and a specificity of 82%. For the premenopausal group, the optimal cutoff value was 23.5, with a sensitivity of 76.5% and a specificity of 62.6%. In the remaining groups (those aged 40 years or younger, and those in postmenopause), the optimal cutoff was identified as 23.1 (Table 4). The ROC analysis resulted in an area under the curve of 0.76 (95% confidence interval, 0.71–0.80) (Fig 1). Table 5 shows a comparison of our results with those from previous studies.

## Discussion

In this study, we utilized the Thai version of the FSFI to assess female sexual function and determined a cutoff score of 23.1 for the screening of FSD using the DSM-5-TR criteria, with a median FSFI score of 24.5 (IQR 20.2–28). Multiple studies conducted in Thailand on the sexual function of women using the FSFI demonstrated that an average overall score for sexual function ranging from 20.0 to 25.0 indicates the presence of sexual dysfunction [10, 23, 24]. To the best of our knowledge, this study is the first attempt to identify the most suitable threshold for FSD screening in the Thai population.

Studies conducted in East Asia (including in China, Japan, and Korea) revealed various cutoff scores ranging from 23.45 to 26.5 [6, 7, 17]. These scores closely align with those reported in prior studies conducted across other Asian countries, with cutoff scores in the 21.9–25.0 range (Table 5) [6, 7, 17–22]. Notably, exceptions were found in studies conducted in Malaysia and the Philippines [25, 26], where a cumulative score without a fixed multiplier for each

**Table 3. Female Sexual Function Index domain score (N = 500).**

| Domain score | Women without FSD (n = 299) Median (IQR) | Women with FSD (n = 201) Median (IQR) | P value |
|---|---|---|---|
| Desire | 3 (2.4–3.6) | 2.4 (1.2–3) | < 0.001 |
| Arousal | 3.9 (3.3–4.5) | 2.7 (1.5–3.6) | < 0.001 |
| Lubrication | 4.8 (3.9–5.6) | 3.6 (2.1–4.8) | < 0.001 |
| Orgasm | 4.8 (4–5.2) | 3.6 (2–4.8) | < 0.001 |
| Satisfaction | 4.8 (3.6–5.6) | 3.6 (3.2–4.8) | < 0.001 |
| Pain | 4.8 (4.4–6) | 4.4 (3.6–5.2) | < 0.001 |
| Total | 26.3 (23.2–29) | 20.8 (14.4–24.9) | < 0.001 |

FSD, female sexual dysfunction; IQR, interquartile range.

**Table 4. Cutoff values for total FSFI score discriminated by different characteristics.**

| Characteristic | Cutoff score | Sensitivity | Specificity | PPV | NPV | LR+ | LR- | AUC |
|---|---|---|---|---|---|---|---|---|
| **All participants** | **23.1** | **75.6** | **67.7** | **77.7** | **65.1** | **2.3** | **0.4** | **0.76** |
| Age ≤40 years | 23.1 | 83.3 | 54.4 | 79.1 | 61.2 | 1.8 | 0.3 | 0.73 |
| Age >40 years | 24.1 | 59.3 | 82.0 | 77.0 | 66.4 | 3.2 | 0.5 | 0.75 |
| Premenopause | 23.5 | 76.5 | 62.9 | 79.2 | 59.2 | 2.1 | 0.4 | 0.74 |
| Postmenopause | 23.1 | 48.6 | 86.2 | 68.0 | 73.5 | 3.5 | 0.6 | 0.69 |

PPV, positive predictive value; NPV, negative predictive value; LR+, positive likelihood ratio. LR-, negative likelihood ratio; AUC: area under the curve.

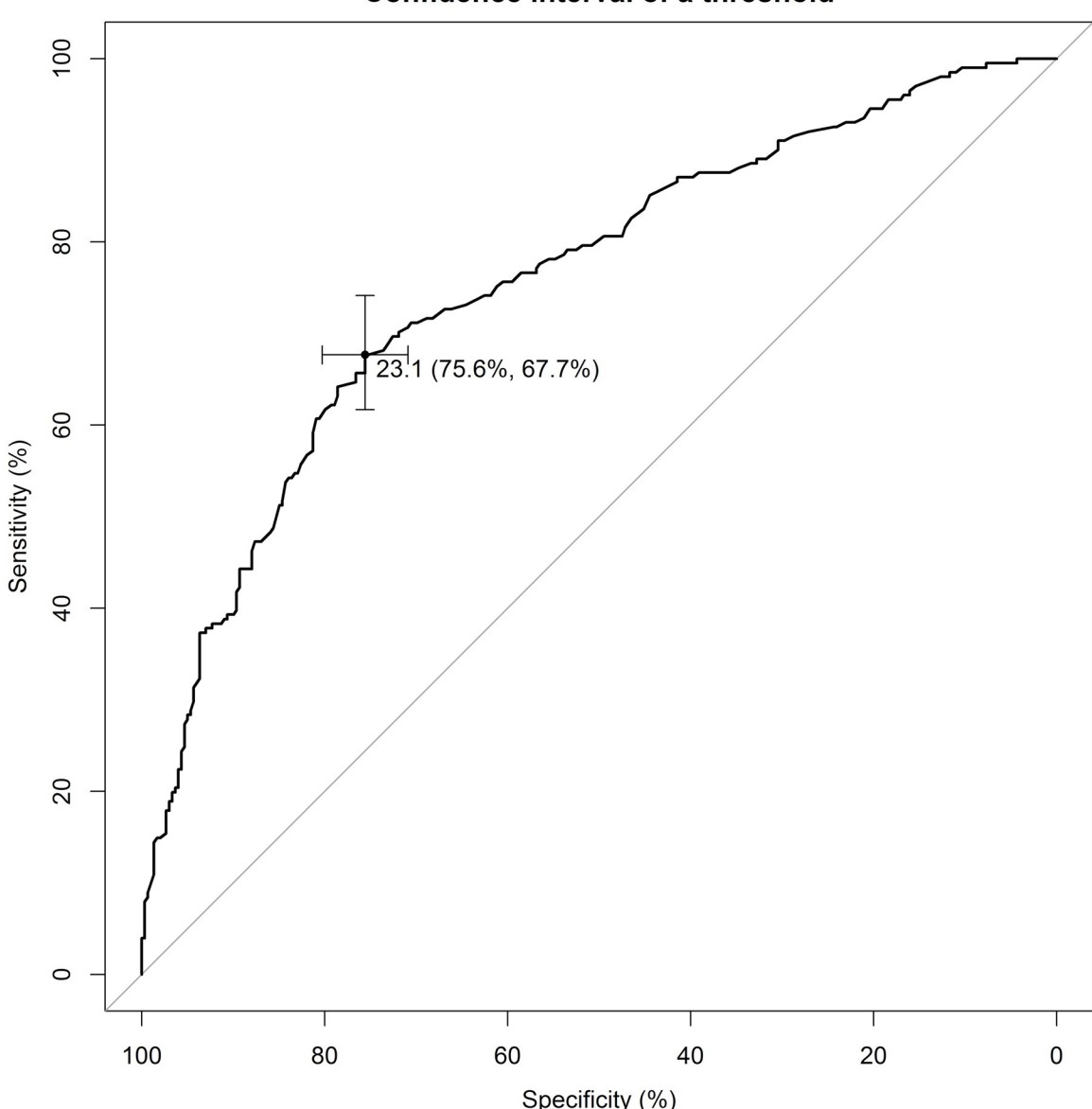

**Fig 1. Receiving operating characteristic curves of the Female Sexual Function Index for screening female sexual dysfunction using Diagnostic and Statistical Manual of Mental Disorders, Fifth Edition, Text Revision criteria.**

**Table 5. Cutoff values for the Female Sexual Function Index in previous studies.**

| Country | Age (years) | No. of participants | Year | Prevalence (%) | Cutoff value | AUC | Sensitivity | Specificity |
|---|---|---|---|---|---|---|---|---|
| Korea (Song et al. [6]) | 18–52 | 504 | 2008 | 43.5 | 25 | 0.83 | 0.66 | 0.89 |
| China (Ma et al. [7]) | 22–60 | 586 | 2014 | 37.6 | 23.45 | 0.75 | 0.66 | 0.72 |
| Japan (Takahashi et al. [17]) | 20–68 | 126 | 2011 | N/A | 23.8 | N/A | N/A | N/A |
| Turkey (Çayan et al. [18]) | 18–66 | 179 | 2004 | 46.9 | 22.7 | N/A | N/A | N/A |
| Bangladesh (Amin et al. [19]) | 45–55 | 260 | 2022 | 56.9 | 21.95 | 0.93 | 0.86 | 0.77 |
| Iran (Safarinejad [20]) | 20–60 | 2,626 | 2006 | 31.5 | 23 | 0.91 | 0.82 | 0.86 |
| Egypt (Ibrahim, Ahmed, and Sayed Ahmed [21]) | 20–59 | 509 | 2013 | 52.8 | 26.55 | 0.99 | 0.97 | 0.93 |
| Colombia (Rincón-Hernández et al. [15]) | 18–52 | 185 | 2021 | 31.4 | 26 | 0.93 | 0.87 | 0.83 |
| Poland (Nowosielski, Wróbel, Sioma-Markowska, and Poręba [22]) | 18–55 | 300 | 2011 | 44.9 | 27.5 | 0.93 | 0.87 | 0.83 |
| United States (Wiegel, Meston, and Rosen [11]) | 18–74 | 568 | 2005 | 54.0 | 26.5 | 0.90 | 0.88 | 0.71 |
| **Thailand (Present study)** | **19–69** | **500** | **2023** | **40.2** | **23.1** | **0.76** | **0.75** | **0.67** |

domain was utilized. Studies in North American and European nations demonstrated comparatively higher scores, ranging from 26.0 to 27.5 [11, 15, 22].

The new DSM-5-TR criteria combine arousal and desire issues into a single disorder, requiring for a longer duration of symptoms, for the symptoms to be experienced on almost all or all occasions, and for the presence of clinically significant distress. In the diagnosis using FSFI scores based on DSM-5-TR, lower scores may indicate patient distress. Consequently, the values are lower compared to scores based on DSM-IV-TR. Cultural variations in symptom acceptance may also influence cutoff scores. The FSFI lacks an assessment of distress required for diagnosing sexual dysfunction, which limits its ability to distinguish between specific domains of FSD. The estimated cutoff point can only determine the possibility of the presence of sexual dysfunction according to DSM-5-TR criteria. Notably, the distress score is an overall measure and does not distinguish between each specific domain/category of FSD. Assessing distress for each domain is crucial, particularly in Eastern cultures, in which FSD may not cause explicit distress. In this cultural background, sex is often viewed as an obligation rather than a source of enjoyment, and as long as women can function well in other areas, complaints of FSD may not significantly impact their overall quality of life.

FSD has a reported global prevalence ranging from 30% to 70%. The prevalence reported in this study closely aligns with that previously reported for other Asian countries. Over the past 15 years, the prevalence of FSD in Thailand among specific groups such as postmenopausal women, those using various types of contraception, and patients with psychiatric disorders ranged from 53% to 96.5% [10, 17, 24, 27–29]. These studies may have involved participants more likely to have FSD. In contrast, this study focused on the general population, and observed a prevalence for the syndrome of 40.2% using DSM-5-TR criteria, consistent with reports from other Asian countries [6, 7, 17].

The prevalence of FSD increases with age, as it is evident in the present study: at least one FSD was identified in 13.9% of individuals aged <29 years, and this percentage increased to 28.4% in the >50 years age group, consistent with findings from previous studies conducted in Turkey, Iran, and Italy [20, 30, 31]. In Turkish women, the prevalence of sexual dysfunction increased from 41% in the 18–30 years age group to 53.1% and 67.9% in the 31–45 and 46–55 years age groups, respectively (FSFI score <25) [30]. Similar patterns were observed in a population-based study in Iran, where the prevalence of FSD increased from 26% in women aged 20–39 years to 39% in women aged >50 years (FSFI score <23) [20]. In Italy, a study

demonstrated a gradual decline in sexual function with age, particularly after 30 and 60 years of age (score <23.4) [31]. The peak in the incidence of FSD occurs between the ages of 40 and 49 years, gradually decreasing thereafter. Sexual dysfunction increases with age in middle-aged men and women [32]. This may be attributed to both individual- and partner-related causes, consistent with previous studies that indicated a decline in sexual function associated with menopausal transition [33]. Other potential factors, in addition to hormonal influences, include physical health, the influence of the partner, the partner's sexual health, and the duration of the relationship [34].

The most prevalent FSD subtype in our study was GPPPD, and this is probably attributable to the influence of physical and emotional symptoms. Based on the DSM-5-TR criteria, we observed a prevalence of 70.6% for GPPPD, followed by FOD at 67.1%, and FSAID at 59.7%. These findings diverged from those of a previous study from Thailand [29], which assessed the prevalence of sexual dysfunction according to DSM-5 and observed at least one sexual dysfunction disorder in 53.3% of the participants: FSAID in 38.0%, FOD in 34.8%, and GPPPD in 6.5%. This discrepancy may arise from the fact that our study focused on healthy individuals, whereas the previous study included participants with psychiatric disorders. Consequently, sexual dysfunction might have been more difficult to detect, particularly considering that most patients in the study were single and may have exhibited fewer signs of sexual dysfunction compared to those in committed relationships.

In a study on women living in Santiago de Chile, 38% reported desire disorders, 32% arousal disorders, 25% orgasmic disorders, 33% dyspareunia [35]. Another study from Brazil reported FSD in 49% of women, sexual desire issues in 26.7%, pain during sexual intercourse in 23%, and orgasmic dysfunction in 21% [36]. Notably, our results are not entirely consistent with the types and numbers reported in the aforementioned studies [37]. Considering the diversity of sexual norms, ranging from gender equality to male-centric attitudes, notably in Asia, there are individual risks and protective variables for sexual dysfunction. In Asia, socio-psychological human development circumstances such as the need to share a bedroom with individuals other than the spouse, delayed initiation of sexual activity, fear of expressing one's sexuality, or the health status of the partner, could affect this condition [38].

In Asia and neighboring countries, cross-sectional surveys have assessed FSD prevalence. In Hong Kong [39], a survey conducted in family planning and pre-pregnancy checkup clinics revealed a 40% prevalence of orgasmic problems, whereas 33.8% of individuals reported pain-related concerns and 31% reported desire and arousal issues, with results comparable to findings from Chile and Brazil. Conversely, in India [40], 77% of the women expressed complaints about arousal disorders, 63% reported desire dysfunction, and 56% reported pain problem, in agreement with the results of our study.

With a distribution that did not distinctly favor any of the three subtypes—GPPPD, FOD, and FSAID—there was no significant difference among these subtypes when comparing Thailand with Hong Kong and India. When comparing the incidence of each subtype, Hong Kong exhibited fewer incidents of all FSD subtypes than Thailand and India. The Hong Kong study primarily emphasized a population of healthy young women with a mean age of 29 years, whereas the Thai and Indian studies focused on an older demographic. It is plausible that the older age in participants from Thailand and India, along with cultural differences affecting the reporting, may lead to an increased reporting of symptoms and to multiple symptom reported by each individual, potentially contributing to a higher prevalence of each disorder.

The attitude toward having self-sexual dysfunction was observed in 29% of the participants. Among them, 60.7% reported desire disorders, 30.3% lubrication disorders, and 28.9% experienced pain problems. Individuals who perceived themselves as having sexual dysfunction were significantly more likely to have at least one sexual dysfunction disorder according to the

DSM-5-TR criteria. This aligns with a study on brief self-assessment of sexual problems conducted in the United States [41], which recommended the use of checklist screens to report clinical tests, emphasizing their specificity and ability to identify specific problems associated with decreased sexual function.

Although only 145 individuals self-reported an attitude toward having self-sexual dysfunction, the structured interviews following the DSM-5-TR framework yielded a higher number of individuals with this attitude. This difference could be attributed to the interactive nature of the interviews, which encompassed open discussions, body language, and tone of voice, potentially prompting patients to report sexual dysfunction more frequently.

The strengths of this study include the use of a validated FSFI screening questionnaire and adherence to the gold standard DSM-5-TR criteria, providing practical guidance for future FSD screening. Similar psychometric measures, such as the Arizona Sexual Experiences Scale (ASEX) [42] and the Pelvic Organ Prolapse/Urinary Incontinence Sexual Questionnaire, IUGA-Revised (PISQ-IR) [43], have also been validated for the Thai population. However, the 5-items scale of ASEX assesses sexual experience in both sexes, not exclusively in women, and is not representative of the general population due to the majority of participants having mental diseases and Parkinson's. Additionally, PISQ-IR is condition-specific, focusing on minimally important differences in symptoms. Therefore, the Thai version of the FSFI appears to be a suitable as a screener for this measure, as it is tailored to the specific language and cultural context.

However, this study has limitations, including a cutoff score of 23.1, with a sensitivity of 75.6% and a specificity of 67.7%. We recommend using the Female Sexual Distress Scale (FSDS) [44] as a secondary screener to assess sexual distress. The absence of a Thai version of the FSDS emphasizes the need for further studies to confirm its effectiveness. Additionally, the lack of predetermined criteria for findings applicable to the general population may introduce subjectivity in the determination of thresholds, potentially impacting generalizability. The personal nature of sexual issues in Thai culture could affect the reproducibility of our findings. Future studies with established cutoff score criteria would enhance the robustness and comparability of the results.

In summary, our findings suggest that FSD is mostly age-related, specifically postmenopausal, with a prevalence above 60% in the age group of over 50 years. The most common subtype is GPPPD, possibly due to vulvovaginal atrophy, which is frequently reported among the symptoms. Seeking further treatment for FSD is a crucial aspect that requires attention. Current evidence indicates that treatment is multimodal and patient-specific, involving strategies such as physical therapy, medical therapy, and lubricants. In recent clinical research settings, the use of energy-based devices resulted in high levels of satisfaction [4, 45, 46]. All these interventions require a multidisciplinary team to improve the quality of life of the women affected by this condition.

## Conclusions

Our study showed that using a cutoff FSFI score of 23.1 for FSD screening yielded a sensitivity of 75.6%, a specificity of 67.7%, and an area under the curve of 0.76 (95% confidence interval, 0.70–0.80). Additionally, we observed an FSD prevalence of 40.2% in the Thai population according to DSM-5-TR criteria.

## Supporting information

**S1 Data.**
(CSV)

## Acknowledgments

The authors express their gratitude to Ms. Jirawan Jayuphan, a statistician in the Department of Epidemiology, Faculty of Medicine, Prince of Songkla University, for providing valuable biostatistical consultations.

## Author Contributions

**Conceptualization:** Patthamaphorn Chongcharoen, Siwatchaya Khanuengkitkong.

**Data curation:** Patthamaphorn Chongcharoen.

**Formal analysis:** Patthamaphorn Chongcharoen, Siwatchaya Khanuengkitkong.

**Investigation:** Patthamaphorn Chongcharoen, Thanapan Choobun, Siwatchaya Khanuengkitkong.

**Methodology:** Patthamaphorn Chongcharoen, Thanapan Choobun, Siwatchaya Khanuengkitkong.

**Project administration:** Patthamaphorn Chongcharoen, Siwatchaya Khanuengkitkong.

**Resources:** Patthamaphorn Chongcharoen, Thanapan Choobun.

**Software:** Patthamaphorn Chongcharoen.

**Supervision:** Siwatchaya Khanuengkitkong.

**Validation:** Thanapan Choobun.

**Visualization:** Thanapan Choobun.

**Writing – original draft:** Patthamaphorn Chongcharoen.

**Writing – review & editing:** Patthamaphorn Chongcharoen, Thanapan Choobun, Siwatchaya Khanuengkitkong.

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
