## [Decision Letter · Decision Letter 0]

26 Dec 2023

PONE-D-23-39011Female sexual function index for screening of female sexual dysfunction using DSM-5-TR criteria in Thai women: A prospective cross-sectional diagnostic studyPLOS ONE

Dear Dr. Khanuengkitkong,

Thank you for submitting your manuscript to PLOS ONE. After careful consideration, we feel that it has merit but does not fully meet PLOS ONE’s publication criteria as it currently stands. Therefore, we invite you to submit a revised version of the manuscript that addresses the points raised during the review process.

We look forward to receiving your revised manuscript.

Kind regards,

Violante Di Donato, Ph.D,M.D.

Academic Editor

PLOS ONE

Journal Requirements:

3. Thank you for stating the following financial disclosure: "This work was supported by the Faculty of Medicine, Prince of Songkla University."

4. We note that your Data Availability Statement is currently as follows: "All relevant data are within the manuscript and its Supporting Information files."

Additional Editor Comments:

Dear Khanuengkitkong

the topic of the present article titled “Female sexual function index for screening of female sexual dysfunction using DSM-5-TR criteria in Thai women: A prospective cross-sectional diagnostic study" is very interesting, the paper and the aim falls within the scope of the journal but the article needs major improvements.

The introduction, material and method section and tables should be modified and improved.

The manuscript should be organized better and English should be improved.

I suggest improving the manuscript with the reviewers' comments.

Reviewers' comments:

Reviewer's Responses to Questions

**Comments to the Author**

1. Is the manuscript technically sound, and do the data support the conclusions?

Reviewer #1: Yes

Reviewer #2: Yes

Reviewer #3: Partly

Reviewer #4: Yes

2. Has the statistical analysis been performed appropriately and rigorously? 

Reviewer #1: Yes

Reviewer #2: Yes

Reviewer #3: Yes

Reviewer #4: Yes

3. Have the authors made all data underlying the findings in their manuscript fully available?

Reviewer #1: Yes

Reviewer #2: Yes

Reviewer #3: Yes

Reviewer #4: Yes

4. Is the manuscript presented in an intelligible fashion and written in standard English?

Reviewer #1: Yes

Reviewer #2: Yes

Reviewer #3: Yes

Reviewer #4: Yes

5. Review Comments to the Author

Reviewer #1: This is a very interesting article covering female sexual dysfunction in an Asian country which is highly likely to be affected with eastern culture.

The Introduction have addressed nicely the novelty of this study.

There are a few points that need to be addressed:

1. The exclusion criteria includes having a partner with sexual dysfunction. Was it all types and degree of sexual dysfunction? How many subjects were excluded due to this criteria?

Question 10 of the questionnaire assessed partner information. Please clarify what kind of information.

Sexual partner's dysfunction may affect the female sexual function and vice versa. The data on the partner's sexual function and how it differed from the FSD group and non FSD group might be interesting to reveal.

2. The distress score is actually an overall score and did not distinguish each domain/category of FSD. The authors have addressed this as a limitation of this study.

Distress score for each domain is important because in eastern cultures many women are experiencing FSD, however it does not actually causing them distress because sex usually is a taboo matter, it is considered only an obligation to be fulfilled instead of being enjoyed and as long as the women can function in other areas, complaints of FSD are not considered something that would affect their overall quality of life. This should be mentioned in the discussion.

3. The prevalence of GPPD, FOD and FSAID in this study is higher compared to previous studies done in Chile and Brazil. The authors should also compare with the data form eastern/neighboring countries of Thailand and elaborate more on the discrepancies if any.

Reviewer #2: The manuscript authored by dr Patthamaphorn Chongcharoen and colleagues aim to define the optimal Thai FSFI cutoff point for screening FSD using DSM-5-TR criteria in Thai women. Also, a second objective was to determine the prevalence of FSD in Thai women using DSM-5-TR criteria

The article is good and the chapters (Introduction, Material and Methods, Results, Discussions and Conclusions) are well written.

The study is a prospective cross-sectional diagnostic study which include a significant number of 500 sexually active women. Each women was evaluated by FSFI and a semi-structured interview about their overall distress defined by DSM-5-TR criteria.

The results are clear presented in 4 tables and compared in table 5 with resulte from other studies.

The discussions are complete.

I propose to accept the article for publication without changes

Reviewer #3: Thank you for the privilege of reviewing this manuscript. This manuscript reports on a validation study of the Thai version of the FSFI to assess female sexual function. Overall, this paper is well-written; a few minor edits are suggested below, given the lack of a priori criteria.

Table 4

1. Cutoff scores of 23.1, 24.1, and 23.5 are listed, but not for the same groups of participants. Please include all observed cutoff scores for: all participants, age ≤40 years, age > 40 years, premenopause, and postmenopause. It is difficult to see the benefits and limitations of the listed cutoff scores because information is omitted (e.g., women >40 years with a cutoff of 23.1, etc.).

Discussion

A cutoff score of 23.1 for screening of FSD using the DSM-5-TR criteria was considered best; however, this resulted in a sensitivity of 75.6 and specificity of 67.7, which seem a bit low. Nearly a quarter of cases are being missed with a sensitivity of only 75.6, and a third of identified cases are incorrectly identified (specificity of 67.7).

2. Please suggest secondary screeners or alternative approaches to utilizing this screener, given the low sensitivity and specificity values.

3. Please include as a limitation the lack of predetermined criteria to identify appropriate cutoff scores.

4. Please reference other measures with similar psychometric properties given the utilized cutoff score; if other measures do not use a cutoff score with 76% sensitivity and 68% specificity, please justify why a cutoff score of 23.1 is suitable as a screener for this particular measure.

Reviewer #4: I read with great interest the Manuscript titled “Female sexual function index for screening of female sexual dysfunction using DSM-5- TR criteria in Thai women: A prospective cross-sectional diagnostic study “. In my opinion, this topic analyzed is interesting enough to attract readers’ attention.

Although the manuscript can be considered already of good quality, I would suggest following recommendations:

- I suggest a round of language revision, in order to correct few typos and improve readability.

- The authors could extend and improved the discussion by evaluating and citing current evidence about possible therapeutic strategy to improve quality of life and symptoms of female sexual functions. I would be glad if the authors discuss this important point, referring to: PMID 32252962 and 36037664.

Because of these reasons, the article should be revised and completed. Considering all these points, I think it could be of interest to the readers and, in my opinion, it deserves the priority to be published after minor revisions.

6. PLOS authors have the option to publish the peer review history of their article (what does this mean?). If published, this will include your full peer review and any attached files.

Reviewer #1: No

Reviewer #2: No

Reviewer #3: No

Reviewer #4: No

---

## [Author Response · Author response to Decision Letter 0]

14 Jan 2024

Editor’s comment

Response: Thank you for your prompt response and valuable feedback.

We have ensured that our revised manuscript aligns with these requirements and have made the necessary changes to meet PLOS ONE's standards. Thank you once again for your guidance and support.

(2) We note that the grant information you provided in the ‘Funding Information’ and ‘Financial Disclosure’ sections do not match. 

Response: Thank you for bringing this to our attention. We appreciate your thorough review, and acknowledge the discrepancy between the grant information provided in the 'Funding Information' and in the 'Financial Disclosure' sections.

In the revised submission, we have ensure that the correct grant numbers for the awards received for our study have been accurately reflected in the 'Funding Information' section. We apologize for any confusion caused by this oversight, and appreciate your diligence in ensuring the accuracy of our manuscript.

Funding: The research was financially supported by the Faculty of Medicine, Prince of Songkla University. Grant 65-086-1, PI: P. Chongcharoen. The funders had no role in the study design, in the data collection and analysis, in the decision to publish, or in the preparation of the manuscript.

Competing interests: The authors have declared that no competing interests exist.

(4) We note that your Data Availability Statement is currently as follows: "All relevant data are within the manuscript and its Supporting Information files."

Response : Thank you for bringing this to our attention. We acknowledge the note regarding our Data Availability Statement, which currently states: "All relevant data are within the manuscript and its Supporting Information files."

Upon your recommendation, we will submit the minimal dataset required to replicate all study findings reported in the article.

Response: Due to the retraction of the paper, the citation to the following article:

Mercer CH, Fenton KA, Johnson AM, Wellings K, Macdowall W, McManus S, et al. Sexual function problems and help seeking behaviour in Britain: National probability sample survey. BMJ. 2003;327: 426–427. doi: 10.1136/bmj.327.7412.426 

has been removed. Instead, a new citation (Reference 5) has been added to the revised manuscript for the following study: 

Amidu N, Owiredu WKBA, Woode E, Addai-Mensah O, Quaye L, Alhassan A, et al. Incidence of sexual dysfunction: a prospective survey in Ghanaian females. Reprod Biol Endocrinol. 2010;8:106. doi: 10.1186/1477-7827-8-106.

 

Reviewer 1

(1) The exclusion criteria includes having a partner with sexual dysfunction. Was it all types and degree of sexual dysfunction? How many subjects were excluded due to this criteria?

Question 10 of the questionnaire assessed partner information. Please clarify what kind of information.

Sexual partner's dysfunction may affect the female sexual function and vice versa. The data on the partner's sexual function and how it differed from the FSD group and non FSD group might be interesting to reveal.

Response: In answer to your question, only patients who met the inclusion criteria but not the exclusion criteria were included. A few prospective participants were excluded after an interview in which the presence of male hypoactive sexual desire disorder, premature ejaculation, delayed ejaculation, or erectile dysfunction (in any degree) in the sexual partner was assessed. Question 10 of the questionnaire specifically assessed irregular cohabitation with partners, which could have a negative impact in terms of stress levels and the frequency of sexual activity. The results revealed no significant effect of irregular cohabitation with partners. In our revised manuscript, we have added a detailed description of Question 10 in the second paragraph of the Materials and Methods section.

(2) The distress score is actually an overall score and did not distinguish each domain/category of FSD. The authors have addressed this as a limitation of this study.

Distress score for each domain is important because in eastern cultures many women are experiencing FSD, however it does not actually causing them distress because sex usually is a taboo matter, it is considered only an obligation to be fulfilled instead of being enjoyed and as long as the women can function in other areas, complaints of FSD are not considered something that would affect their overall quality of life. This should be mentioned in the discussion.

The distress score is actually an overall score and did not distinguish each domain/category of FSD. The authors have addressed this as a limitation of this study.

Response: We appreciate the reviewer's insightful comments regarding the distress score in our study. We acknowledge that the distress score provided in our analysis is an overall score and does not distinguish between domains/categories of FSD. As rightly pointed out, this limitation has been addressed in the Discussion section of our revised manuscript (third paragraph).

(3) The prevalence of GPPPD, FOD and FSAID in this study is higher compared to previous studies done in Chile and Brazil. The authors should also compare with the data form eastern/neighboring countries of Thailand and elaborate more on the discrepancies if any.

Response: We appreciate the reviewer's observation regarding the prevalence of GPPPD, FOD, and FSAID in our study compared to previous research conducted in Chile and Brazil. We acknowledge the importance of providing a comprehensive analysis, including a comparison with data from other Eastern countries, including those neighboring to Thailand. In our revised manuscript, we have incorporated a detailed comparison with relevant studies from this regions to elucidate the variations observed in the prevalence of GPPPD, FOD, and FSAID. This comparison can be found in the eighth paragraph of the Discussion section.

 

Reviewer 2

The manuscript authored by Dr Patthamaphorn Chongcharoen and colleagues aim to define the optimal Thai FSFI cutoff point for screening FSD using DSM-5-TR criteria in Thai women. Also, a second objective was to determine the prevalence of FSD in Thai women using DSM-5-TR criteria

The article is good and the chapters (Introduction, Material and Methods, Results, Discussions and Conclusions) are well written.

The study is a prospective cross-sectional diagnostic study which include a significant number of 500 sexually active women. Each women was evaluated by FSFI and a semi-structured interview about their overall distress defined by DSM-5-TR criteria.

The results are clear presented in 4 tables and compared in table 5 with result from other studies.

The discussions are complete.

I propose to accept the article for publication without changes.

Response: 

Thank you very much for your positive feedback and thorough review of our manuscript. We appreciate your kind remarks regarding the clarity and structure of the manuscript sections. We are particularly glad that the results, presented in tables and compared in Table 5 with findings from other studies, have met your expectations.

Your acknowledgment of the completeness of the Discussion section is encouraging, and we are pleased that the study design and methods were well-received. We are grateful for your suggestion to accept the article for publication without changes. Thank you for your time and valuable insight.

 

Reviewer 3

(1) Cutoff scores of 23.1, 24.1, and 23.5 are listed, but not for the same groups of participants. Please include all observed cutoff scores for: all participants, age ≤40 years, age > 40 years, premenopause, and postmenopause. It is difficult to see the benefits and limitations of the listed cutoff scores because information is omitted (e.g., women >40 years with a cutoff of 23.1, etc.).

A cutoff score of 23.1 for screening of FSD using the DSM-5-TR criteria was considered best; however, this resulted in a sensitivity of 75.6 and specificity of 67.7, which seem a bit low. Nearly a quarter of cases are being missed with a sensitivity of only 75.6, and a third of identified cases are incorrectly identified (specificity of 67.7).

Response: We conducted subgroup analyses based on cutoff values derived from the total FSFI score and various characteristics. Our findings revealed that among women aged over 40 years, the optimal cutoff was 24.1, demonstrating a sensitivity of 59.3% and a specificity of 82%. For the premenopausal group, the optimal cutoff was 23.5, with a sensitivity of 76.5% and a specificity of 62.6%. Additionally, in other groups, such as those aged 40 years or younger and those in postmenopause, the optimal cutoff was identified as 23.1 (Table 4). We have included these optimal cutoff values in the fifth paragraph of the Results section in our revised manuscript.

The data we analyzed is shown in the table below.

Character Cutoff score Sensitivity Specificity PPV NPV LR+ LR- AUC

All participants 23.1 75.6 67.7 77.7 65.1 2.3 0.4 0.76

 23.5 72.6 69.7 78.1 63.1 2.4 0.4 0.76

 24.1 68.2 71.6 78.2 60.3 2.4 0.4 0.76

Character Cutoff score Sensitivity Specificity PPV NPV LR+ LR- AUC

Age ≤40 years 23.1 83.3 54.4 79.1 61.2 1.8 0.3 0.73

 23.5 79.6 57.8 79.6 57.8 1.9 0.4 0.73

 24.1 73.7 58.9 78.7 52.0 0.8 0.4 0.73

Character Cutoff score Sensitivity Specificity PPV NPV LR+ LR- AUC

Age >40 years 23.1 62.8 78.4 74.7 67.4 2.9 0.8 0.75

 23.5 61.1 79.3 75.0 66.7 2.9 0.5 0.75

 24.1 59.3 82.0 77.0 66.4 3.2 0.5 0.75

Character Cutoff score Sensitivity Specificity PPV NPV LR+ LR- AUC

Premenopause 23.1 79.2 60.1 78.6 61.0 2.0 0.3 0.74

 23.5 76.5 62.9 79.2 59.2 2.1 0.4 0.74

 24.1 71.6 65.7 79.4 55.6 2.1 0.4 0.74

Character Cutoff score Sensitivity Specificity PPV NPV LR+ LR- AUC

Postmenopause 23.1 48.6 86.2 68.0 73.5 3.5 0.6 0.69

 23.5 42.9 86.2 65.2 71.4 3.1 0.7 0.69

 24.1 42.9 86.2 65.2 71.4 3.1 0.7 0.69

(2) Please suggest secondary screeners or alternative approaches to utilizing this screener, given the low sensitivity and specificity values.

Response: We appreciate the reviewer's insightful comment regarding the low sensitivity and specificity values observed in our screener. Recognizing the importance of improving the screening tool, we are committed to addressing this limitation. In our revised manuscript, we explore and discuss potential secondary screeners or alternative approaches, specifically in the eleventh and twelfth paragraph of the revised Discussion section.

(3) Please include as a limitation the lack of predetermined criteria to identify appropriate cutoff scores.

Response: We appreciate the reviewer's insightful comment regarding the lack of predetermined criteria to identify appropriate cutoff scores in our study. Recognizing the importance of transparency, we have included this as a limitation in our revised manuscript, specifically in the twelfth paragraph of the Discussion section.

(4) Please reference other measures with similar psychometric properties given the utilized cutoff score; if other measures do not use a cutoff score with 76% sensitivity and 68% specificity, please justify why a cutoff score of 23.1 is suitable as a screener for this particular measure.

Response: We appreciate the reviewer's suggestion. In the eleventh paragraph of the revised Discussion section, we have thoroughly discussed this point and included relevant references to measures that align with these criteria.

Reviewer 4

(1) I suggest a round of language revision, in order to correct few typos and improve readability.

Response: Thank you for your constructive feedback. We appreciate your suggestion for a round of language revision to correct typos and improve readability. We have undertaken a thorough review of the text and made the necessary revisions to enhance the overall clarity and readability of the manuscript. Your input is valuable to us. Once again, thank you for your time and thoughtful recommendations.

(2) The authors could extend and improved the discussion by evaluating and citing current evidence about possible therapeutic strategy to improve quality of life and symptoms of female sexual functions. I would be glad if the authors discuss this important point, referring to: PMID 32252962 and 36037664.

Response: We sincerely appreciate the reviewer's insightful comment and guidance. Recognizing the significance of discussing therapeutic strategies to enhance the quality of life and address the symptoms of female sexual dysfunction, we have extended and improved the Discussion section in our revised manuscript. We specifically refer to the provided PMID 32252962 and 36037664 and incorporated the current evidence related to potential therapeutic interventions. Thank you for pointing us towards these relevant references. We have incorporated this additional information into the thirteenth paragraph of our revised Discussion.

---

## [Decision Letter · Decision Letter 1]

2 Feb 2024

Female sexual function index for screening of female sexual dysfunction using DSM-5-TR criteria in Thai women: A prospective cross-sectional diagnostic study

PONE-D-23-39011R1

Dear Dr. Khanuengkitkong,

We’re pleased to inform you that your manuscript has been judged scientifically suitable for publication and will be formally accepted for publication once it meets all outstanding technical requirements.

Kind regards,

Violante Di Donato, Ph.D,M.D.

Academic Editor

PLOS ONE

Additional Editor Comments (optional):

The manuscript has been modified with the comments of the reviewers. It is now ready to be published.

Reviewers' comments:

Reviewer's Responses to Questions

**Comments to the Author**

1. If the authors have adequately addressed your comments raised in a previous round of review and you feel that this manuscript is now acceptable for publication, you may indicate that here to bypass the “Comments to the Author” section, enter your conflict of interest statement in the “Confidential to Editor” section, and submit your "Accept" recommendation.

Reviewer #1: All comments have been addressed

Reviewer #2: All comments have been addressed

Reviewer #3: All comments have been addressed

2. Is the manuscript technically sound, and do the data support the conclusions?

Reviewer #1: Yes

Reviewer #2: Yes

Reviewer #3: Yes

3. Has the statistical analysis been performed appropriately and rigorously? 

Reviewer #1: Yes

Reviewer #2: Yes

Reviewer #3: Yes

4. Have the authors made all data underlying the findings in their manuscript fully available?

Reviewer #1: Yes

Reviewer #2: Yes

Reviewer #3: Yes

5. Is the manuscript presented in an intelligible fashion and written in standard English?

Reviewer #1: Yes

Reviewer #2: Yes

Reviewer #3: Yes

6. Review Comments to the Author

Reviewer #1: All reviewer's comments have been addressed by the authors. I recommend to accept this article to be published.

Reviewer #2: Dear Authiors,

as mentioned at the first review round I agree with publication in this form.

I find your research well done and the questions from the other reviewers also improved the content.

thank you

Reviewer #3: Thank you for the privilege of reviewing this manuscript; all suggested revisions have been adequately made. I have no further reservations towards accepting this submission.

7. PLOS authors have the option to publish the peer review history of their article (what does this mean?). If published, this will include your full peer review and any attached files.

Reviewer #1: No

Reviewer #2: No

Reviewer #3: No

---

## [Editor Report · Acceptance letter]

12 Feb 2024

PONE-D-23-39011R1 

PLOS ONE

Dear Dr. Khanuengkitkong, 

I'm pleased to inform you that your manuscript has been deemed suitable for publication in PLOS ONE. Congratulations! Your manuscript is now being handed over to our production team.

Kind regards, 

on behalf of

Dr. Violante Di Donato 

Academic Editor

PLOS ONE